# Chairman Narcissism and Social Responsibility Choices: The Moderating Role of Analyst Coverage

**DOI:** 10.3390/bs13030245

**Published:** 2023-03-10

**Authors:** Qingzhu Gao, Liangmou Gao, Dengjie Long, Yuege Wang

**Affiliations:** 1School of Business Administration, Dongbei University of Finance and Economics, Dalian 116012, China; 2Party School of Chongqing Committee of C.P.C, Marxism College, Chongqing 400041, China; 3Faculty of Agribusiness and Commerce, Lincoln University, Canterbury 7647, New Zealand

**Keywords:** corporate social responsibility (CSR), CSR strategies, behavioral consistency theory, personality traits, narcissistic chairman, management autonomy, cognitive and motivational factors

## Abstract

Chairman narcissism has received extensive attention in social psychology and organizational behavior, but the relationship between chairman narcissism and social responsibility has not yet received much attention. The purpose of this study is to investigate the effect of chairman narcissism on various dimensions of CSR and the moderating roles of analyst coverages. Based on upper echelons theory and stakeholder theory, we distinguished internal corporate social responsibility (internal CSR) and external corporate social responsibility (external CSR) according to whether there was a formal contractual relationship. This study used a narcissism index of chairmen of Chinese listed companies to examine the relationship between chairman narcissism and internal CSR, external CSR, and the data were analyzed using Stata16.0. The results showed that there was a positive correlation between chairman narcissism and external CSR, and there was a negative correlation between chairman narcissism and internal CSR. That is, the higher the Chairman’s narcissism degree is, the more external CSR and less internal CSR the firm makes. Further research showed that analyst coverage has weakened the impact of chairman narcissism on internal and external CSR. This paper enriches and expands the research on chairman narcissism and CSR and provides new ideas for selecting corporate managers and improving corporate governance.

## 1. Introduction

Corporate social responsibility is the responsibility that firms undertake beyond their interests and legal requirements [1,2]. The external environmental (e.g., legal environment, government response, media attention), the firm itself (e.g., economic performance, innovation capability, ownership structure), and the managerial team’s personality traits and demographic characteristics will affect CSR [3]. As the most powerful and influential individual in the managerial team, the chairman has the most direct and essential impact on CSR [4]. Compared with the demographic characteristics of the chairman, the personality traits of the chairman have a more significant impact on CSR [5,6]. Narcissism is an essential dimension of the chairman’s personality traits, which reflects the personality traits that the chairman overestimates his abilities and charisma, desires to receive higher praise from others, and focuses on realizing self-interest [7,8]. Previous studies have found that the higher chairman’s narcissism degree is, the more he tends to focus on behaviors and issues that can attract external attention [9,10]. As an essential issue that can attract stakeholders’ attention and appreciation, CSR will inevitably be influenced by the chairman’s narcissistic traits.

The influence of chairman narcissism on CSR has received attention from scholars. Based on the behavioral consistency theory, some scholars believe that chairmen with narcissistic traits will show corresponding management styles in operating and managing the firm and that charitable donation is necessary to win stakeholders’ attention. Some scholars, based on the hypothesis of rational man, emphasize that narcissistic chairmen will maximize their interests at the expense of other stakeholders [10,11]. Based on the Attention-based View, some scholars also found that narcissistic leaders tend to take on CSR to satisfy their desire for external attention. However, when the narcissism level exceeds a certain threshold, they will focus on programs that can better satisfy their narcissistic needs [12]. The above studies show that the relationship between chairman narcissism and CSR is inconsistent. There are two possible reasons. First, existing studies focus on one aspect of CSR (e.g., consumer responsibility, government responsibility, employee responsibility) or have yet to classify the types of CSR. Different types of CSR have different implementation cycles and costs, social attention, and recognition, and represented stakeholders. Driven by self-interested needs and beliefs, narcissistic chairmen usually exhibit different behavioral logics. Second, studies have ignored the effect of analyst coverage on the relationship between chairman narcissism and CSR.

The main purpose of the study is to investigate the effect of chairman narcissism on various dimensions of CSR and the moderating role of analyst coverages. To accomplish this, the paper proceeds as follows. First, drawing upon Stakeholder Theory, the target of CSR behavior was identified as the firms’ stakeholders. According to whether stakeholders have a formal market transaction contractual relationship with the firm, this paper specifies internal and external stakeholders [13] and refers to CSR toward external stakeholders as external CSR and CSR toward internal stakeholders as internal CSR. Second, Upper Echelons Theory shows that the psychological structure of the chairman’s cognitive base, values, and psychological preferences are the essential factors that influence decision-making and corporate behavior. Using the sample of listed firms in China from 2009 to 2019, we empirically examine the effect of chairman narcissism on internal and external CSR. Third, considering that analyst coverage may be an essential external governance factor affecting the management autonomy of the chairman, this paper introduces it and examines the moderating effect of analyst coverages.

This paper contributes to the extant literature on chairman narcissism and on internal and external CSR in several ways. First, this study contributes to the growing literature on the relationship between chairman narcissism and the choice of CSR strategies. Our results show that chairman personality traits, especially narcissism, can significantly impact the choice of a firm’s nonmarket strategies in China, an important emerging economy. Compiling a chairman narcissism index from a video survey, we show that firms with narcissistic chairmen engage in more external CSR activities and less internal CSR activities. We provide a more comprehensive assessment of the relationship between chairman narcissism and CSR strategies and clarify the paradoxical relationship between chairman narcissism and CSR. Second, we respond to the growing calls for deconstructing the CSR measure. Our study distinguished CSR into internal CSR and external CSR, this division extending and enriching the literature on internal CSR and external CSR. More importantly, we expand the theory of stakeholder. Meanwhile, by relating chairmen’ psychological characteristics, such as narcissism, to explain CSR choices, we open up doors for future research on the discretionary determinants of CSR. Third, this paper broadens the boundaries of the relationship between chairman narcissism and CSR. Based on the call, narcissistic traits should be studied in the context of “activating” and “inhibiting” [14]. We chose analyst coverage as the moderator variable, and the results show that analyst coverage weakened the relationship between chairman narcissism and CSR, which enriches the contingency perspective of chairman narcissism.

## 2. Literature Review and Hypothesis Development

### 2.1. Chairman Narcissism and Disentangling Corporate Social Responsibility

Narcissism is a fundamental personality trait that refers to an individual’s self-worship, self-superiority, lack of empathy, selfishness, and pursuit of authority, power, and exploitation [15,16]. Unlike overconfidence and self-esteem, narcissistic individuals are characterized by a desire for lasting appreciation. Compared with dark personality traits, such as Machiavellianism, narcissism also exhibits more self-reinforcement and high extraversion [17,18]. As a stable psychological and personality trait, narcissism is prevalent in managers. The reason is that narcissists possess traits, such as self-confidence and leadership, that can help them become managers. Chairman narcissism contains both cognitive and motivational factors. Cognitively, a narcissistic chairman has an inflated self-concept. They are usually self-centered, believe that their decisions are always correct, tend to be arrogant, callous, and like to deny the views of others [19,20]. In terms of motivation, narcissistic chairmen are motivated to seek power, control, and inflated ego, especially motivated by various types of behavior inviting applause and admiration [19,21]. Upper echelon theory emphasizes the chairman’s personality traits and psychological characteristics are essential factors that affect decision-making [22]. As an important dimension of the chairman’s traits, narcissism can reflect the psychological and cognitive characteristics of the chairman and will inevitably affect CSR decisions.

Stakeholder theory argues that the firm can be considered as a collection of contracts composed of stakeholders [23,24]. Those stakeholders who have signed formal market contracts with the firm, such as shareholders, managers, employees, etc., can be called “internal stakeholders”. On the contrary, those stakeholders who have not signed formal market contracts with the firm are called “external stakeholders”, such as the government, media, community, and the public. Drawing upon stakeholder theory, this paper refers to CSR toward external stakeholders as external CSR, which focuses on external charitable donations [25,26]. The CSR toward internal stakeholders is called internal CSR, which focuses on employee welfare investment [27,28]. Internal and external stakeholders have different demands for CSR. Specifically, internal stakeholders demand CSR based on their interests, while external stakeholders emphasize that the CSR undertaken by the firm should be in line with social ethics and norms. In this context, narcissistic chairmen usually show different behavioral logic when considering what kind of CSR to undertake. That is, driven by narcissistic traits, narcissistic chairmen will consider whether undertaking internal CSR can maximize self and corporate interests based on instrumentalism. When undertaking external CSR, narcissistic chairmen prioritize whether they can gain attention and praise from external stakeholders.

### 2.2. Chairman Narcissism and External Corporate Social Responsibility

External stakeholders demand that firms undertake CSR based on the social contract, but firms pay more attention to the economic benefits that result from achieving the social contract. Dyer et al. [29] argue that satisfying the interests of external stakeholders may not enhance the firm’s economic performance, but they can gain positive evaluations. These positive evaluations are the basis for firms to establish image and reputation and are also an important source of “narcissistic supply” for narcissistic chairmen [21,30]. Therefore, narcissistic chairmen are more likely to undertake external CSR to gain the attention, recognition, and praise of external stakeholders.

On the one hand, chairmen with a high degree of narcissism focus on external attention and praise and need to constantly find and replenish “narcissistic supply sources” [21]. Recognition and praise from external stakeholders are an essential source of “narcissistic supply” for narcissistic chairmen. External CSR is an effective way to obtain positive evaluations from external stakeholders [31]. Therefore, narcissistic chairmen are more willing to engage in charitable donations and other external CSR activities to gain lasting appreciation from external stakeholders. Recent research has found that narcissistic chairman’s reputation-seeking personality drives his passion for philanthropy and charitable giving.

On the other hand, chairmen with high narcissism focus on their interests and tend to look for ways to satisfy them, thus increasing the possibility of fulfilling external CSR. Compared to chairmen with low narcissism, chairmen with high narcissism have a strong sense of entitlement and are willing to actively pursue what they believe they deserve [30]. O’Reilly et al. [32] found that the self-interest characteristics of narcissistic CEOs drive them to seek higher compensation. External CSR can significantly improve corporate financial performance by determining the chairman compensation [33]. Thus, increasing external CSR investment may be essential for narcissistic chairmen to increase their compensation. In addition, with the increase of external CSR investment, the resources that the chairmen control will increase, which provides opportunities for narcissistic chairmen to capture private profits. Therefore, we generated the following hypothesis.

**Hypothesis** **1** **(H1).**
*Chairman narcissism positively influences external CSR.*


### 2.3. Chairman Narcissism and Internal Corporate Social Responsibility

Stakeholder theory shows that the company can be regarded as a collection of contractual relationships consisting of internal stakeholders whose survival and growth of the collection depend on the collective contribution of the internal stakeholders [23]. Theoretically, the chairman of the board is also an internal stakeholder. He signs contracts directly or indirectly with internal stakeholders (e.g., employees) on behalf of the company and plays a crucial role in corporate decision-making. Narcissism is an essential dimension of the chairman’s personality. Chairmen with high levels of narcissism are often self-centered, distrusted, callous [34], focused on achieving personal benefits, and demanding of their subordinates, which will eventually inhibit CSR engagement with internal stakeholders.

On the one hand, chairmen with high levels of narcissism have obvious negative traits, such as egoism and intellectual inhibition, which inhibit internal CSR behaviors. Compared to chairmen with low narcissism, chairmen with high narcissism are self-serving and exploitative of internal employees. Such bullying behavior can be seen at work, such as harsh criticism of employees, and stealing their work, which violates their interests [10,35]. At the same time, narcissist chairman also exhibits higher vulnerability, including distrust, deceitfulness, oppositionality, callousness, and anxiety [36,37]. Meanwhile, chairmen with high narcissism focus on maintaining their authority and will maintain it through negative management, such as denying, criticizing, or even insulting employees. In addition, chairmen with high narcissism attribute employee successes to themselves and failures to their employees. Chairmen with high levels of narcissism will ignore or even resent views that do not agree with him and will threaten or even emotionally attack employees who hold different views and negative feedback.

On the other hand, chairmen with high levels of narcissism focus on short-term economic benefits and neglect long-term investment projects, resulting in the suppression of internal CSR. Compared with chairmen with low narcissism, chairmen with high narcissism tend to choose projects that can bring social attention and economic benefits to individuals in the short term. Although internal CSR is embedded in corporate systems, business operations, and daily management, it takes a long time to cultivate and form. The cost of training is relatively high, which makes it difficult for external stakeholders to identify and evaluate the chairmen’s efforts in CSR. On the contrary, external CSR (e.g., charitable donation, environmental investment) has the advantages of being easily visible to the public, having a shorter implementation cycle, and being sustainable [38]. Therefore, narcissistic chairmen are more likely to increase spending related to external CSR and decrease spending related to internal CSR. Therefore, we generated the following hypothesis.

**Hypothesis** **2** **(H2).**
*Chairman narcissism negatively affects internal CSR.*


### 2.4. The Moderating Effect of Analyst Coverage

As an essential information intermediary and external governance mechanism in the capital market, analysts can use their professional information network and expertise to provide the capital market with a large amount of high-quality information [39,40]. Moreover, analysts are related to the market, industry, and firms, which can reduce the information asymmetry between the chairman of the board and investors and shareholders. They can also reduce the chairman’s opportunistic behavior by increasing the capital market’s supervision of the chairman. Qian et al. [40] found that analysts can exert the “information revealing effect” and “monitoring effect” to prevent opportunism and short-sighted behavior of the chairman. This paper predicts that analyst coverage can moderate the relationship between chairman narcissism and external CSR and internal CSR.

First, from the perspective of “information disclosure effect”, analyst coverage can reduce information asymmetry between CSR inputs and stakeholders, improve information transparency, and reduce the impact of irrational narcissistic traits of the chairman on external and internal CSR. Specifically, analysts can summarize information related to chairman narcissism and CSR and submit it to outside investors in the form of published research reports so that outsiders can comprehensively identify the internal and external CSR behavior of narcissistic chairman [41,42]. The higher the transparency of the listed company is, the more likely it is that excessive external charitable donations and suppression of internal employee behavior will be detected. At this point, the irrational behavior of narcissistic chairmen will be reduced for reputation and career development reasons. In addition, the shareholders of a corporation can correctly recognize the irrational CSR behavior of the narcissistic chairmen through the research reports issued by the analysts and put pressure on the narcissistic chairmen out of their interests.

Second, from the perspective of “monitoring effect”, analyst coverage can enhance the monitoring of the capital market on a narcissistic chairman and mitigate the impact of the chairman’s irrational narcissism on external and internal CSR [43,44]. When the narcissistic chairmen unreasonable external CSR and violations of internal employee welfare are discovered and reported by analysts, they will not only be severely punished by the shareholders but will also damage their reputation in the managerial market and affect the career development prospects of the narcissistic chairmen [39,45]. In other words, the external governance role of analysts will promote the convergence of the interests of narcissistic chairmen and shareholders, motivate the narcissistic chairmen to make better internal and external CSR decisions based on the interests of shareholders, and reduce the adverse effects of the irrational narcissistic traits of narcissistic chairmen on external CSR and internal CSR. Therefore, we generated the following hypothesis.

**Hypothesis** **3a** **(H3a).**
*Analyst coverage weakens the positive relationship between chairman narcissism and external CSR.*


**Hypothesis** **3b** **(H3b).**
*Analyst coverage weakens the negative relationship between chairman narcissism and internal CSR.*


## 3. Methods

### 3.1. Sample

Our sample included Chinese A-share listed firms in Shanghai and Shenzhen stock exchanges from 2010 to 2019. The design of sample selection is as follows: Excluding ST, ST*, and PT firms; excluding financial and insurance listed firms; excluding firms whose chairman has served for less than three years; excluding firms listed less than three years; excluding samples with missing variable data. Our final sample comprises 1212 firm-year observations, which contain 169 firms. Data on corporate chairman narcissism were collected from a video survey methodology and videos of chairmen in our sample through publicly available internet sources, such as Baidu.com and hao.360.com search engines. The names of the firms and chairmen’s names were obtained from the Chinese Stock Market Research (CSMAR) database. CSMAR is a leading and professional financial database of Chinese listed firms. Data on charitable donations, chairman characteristics, and corporate characteristics were collected from the Chinese Stock Market Research (CSMAR) database. To reduce the effect of abnormal data on the regression results, 1% winsorize was applied on continuous variables, and Stata16.0 was used for data processing and model estimation.

### 3.2. Variables and Measures

#### 3.2.1. Dependent Variable: Internal and External CSR

Based on stakeholder theory, we divided the service targets of CSR into external and internal stakeholders. External CSR emphasizes that firms should take responsibility for their external stakeholders. Charitable donation is a typical proxy variable of external CSR. Following Lin et al. [46] research, Ecsr (External Corporate Social Responsibility) = Ln (Total annual charitable donations + 1). Internal CSR refers to a company’s responsibility to its internal stakeholders. Employees are the most important internal stakeholders, and the level of welfare and social security provided by the company to employees can measure internal CSR. Drawing on the studies of Beijing Yang and Lu Feng [47], Icsr (Internal Corporate Social Responsibility) = Ln (Employee social insurance expenditures+ employee training expenditures + 1).

#### 3.2.2. Independent Variable: Chairman Narcissism

Chairman narcissism is a psychological state in which the chairman focuses on himself, over-evaluates himself, seeks power, and desires praise from others. Based on the Chinese institutional context, the chairman of the board is the most powerful individual in the managerial team, directly influencing CSR decisions and behaviors. As a public figure, the chairman of the board has a large amount of information on the Internet that can provide a basis for the study of chairman narcissism. Following Petrenko et al. [21], Fung et al. [48], Zhu and Chen. [49], this paper uses a video survey method to study chairman narcissism. Specific steps include.

First, drawing on the narcissistic personality inventory developed by Raskin and Hall [50], Fung et al. [48], Qiao et al. [51], and also based on the narcissistic dimensions delineated by Emmons [52], a narcissism scale applied to the chairman of the board was compiled. Management experts and senior managers were invited to review the narcissism scale, which was modified based on comments. The revised Narcissism Scale for chairman included 16 indicators (NPI-16) in 4 dimensions. The four dimensions include: Entitlement/ exploitativeness; authority/leadership; superiority/arrogance; self-admiration/self-absorption. In addition, the study used a Likert 7-point scale ranging from 1 to 7 (1 = strongly disagree, 7 = strongly agree). The details of the 16-item measure are shown in Table 1.

Second, drawing on Petrenko et al. [21] study, we manually collated the chairmen videos using search engines, such as Baidu and 360, and divided the videos into five groups according to their length (1–3 min, 3–5 min, 5–10 min, 10–30 min, and more than 30 min). To ensure the validity and credibility of the narcissism of the chairman, we set up a “narcissism test” session. In other words, we randomly selected 30 chairmen and examined the ratings of the same chairman in different-length videos. The results showed that the different video duration did not significantly affect the narcissism degree of the same chairman (*p* > 0.1). To avoid the influence of observer fatigue on the evaluation results, we chose 5–10 min of the chairman’s video for research.

Third, we determined the total sample of the score, while developing a rigorous scoring procedure. On the one hand, 12 psychology graduate students with experience in personality assessment were selected as raters. The raters were trained and given pecuniary rewards according to the scoring procedure. On the other hand, we created an excel spreadsheet consisting of the NPI-16 scales and divided the 12 raters into 6 groups. The raters were required to enter the scoring system with an individual login password, watch the video independently within the specified time, and score the narcissism degree of each chairman. The raters also required to enter their ratings and justifications for each chairman’s rating in an Excel spreadsheet. The justifications described the reasons why the rater gave a particular score to a chairman [50]. Specific justifications included whether a chairman used friendly facial expressions, exhibited body posture, spoke casually, and wore colorful clothing [48]. Finally, each chairman was given less than 30 min to score, and all scoring was completed within two weeks.

Fourth, the narcissism results were selected by comparing the score differences. If the difference between the rating scores of two raters in the group was more significant than the standard deviation of the total sample, the other group of raters would be arranged for evaluation. If only one raters’ score was significantly different from others, the raters’ score would be excluded, and the mean of the other three raters’ scores was selected as the narcissism index of the chairman. If the difference between the scores of the four raters was large, other videos of the chairman of the board would be provided for reevaluation.

#### 3.2.3. Moderator Variable: Analyst Coverage

Analysts can restrain the irrational CSR behavior of the narcissistic chairmen through the market pressure mechanism, external governance mechanism, and information intermediary mechanism. Specifically, the higher the analyst coverage of enterprises, the more comprehensive and multidimensional the true picture of CSR will be revealed and interpreted, and the lower the motivation of narcissistic chairman to implement irrational CSR. Referring to the studies of He and Tian [53], analyst coverage is measured by the number of analysts reporting on the firm per year. Ana (Analyst coverage) measurements use the following formula. Ana = Ln (Number of firms covered by analysts each year + 1).

#### 3.2.4. Control Variables

We included control variables at two levels: Chairman characteristics and firm characteristics. First, we controlled for the chairman’s gend, lage, and edu. Gend (chairman’s gender) is a dummy variable, 1 for males and 0 for females. Lage (chairman’s age) is the age of the chairman in years. The Edu (chairman’s degree) variable equals 1 if the chairman has a higher school and below degree, 2 for an associate degree, 3 for a bachelor’s degree, 4 for a master’s degree, 5 for a PhD degree. Second, firm characteristic variables include size, age, lev, dudb, sub, board, con. Size (firm size) was the natural logarithm of the total employees employed. Age (firm age), the number of years the firm has been in business. Lev (firm leverage) was measured as the ratio of total liabilities to total assets. Dudb (independent directors) was measured as the ratio of independent directors to the total number of boards of directors. Sub (government subsidies) was the natural logarithm of government subsidies at the end of the year. Board (board size) was the natural logarithm of the total number of board members. Con (ownership concentration) was measured as the ratio of top ten shareholders to the total share capital.

### 3.3. Model Construction

The following models are established to test Hypothesis 1 and Hypothesis 2.
(1)$${Ecsr}_{i,t}=a_{0}+a_{1}{Nar}_{i,t}+a_{2}Controls+\varepsilon_{i,t}$$
(2)$${Icsr}_{i,t}=\beta_{0}+\beta_{1}{Nar}_{i,t}+\beta_{2}Controls+\varepsilon_{i,t}$$
where Ecsr is external CSR, Icsr is internal CSR, Nar is chairman narcissism, $a_{0}$, $\beta_{0}$ represents the intercept terms, Controls stands for all control variables, ε stands for error terms, i and t represent an individual firm’s observation and each year, respectively. If $a_{1}$ is significantly positive, indicates that Hypothesis 1 is validated. If $\beta_{1}$ is significantly negative, indicates that Hypothesis 2 is validated.
(3)$${Ecsr}_{i,t}=\mu_{0}+\mu_{1}{Nar}_{i,t}+\mu_{2}{Nar}_{i,t}\times A{na}_{i,t}+\mu_{3}A{na}_{i,t}+\mu_{4}Controls+\varepsilon_{i,t}$$
(4)$${Icsr}_{i,t}=\omega_{0}+\omega_{1}{Nar}_{i,t}+\omega_{2}{Nar}_{i,t}\times A{na}_{i,t}+\omega_{3}A{na}_{i,t}+\omega_{4}Controls+\varepsilon_{i,t}$$

Nar×Ana represents the interaction between chairman narcissism and analyst coverage to test for the moderating effect. If $\mu_{2}$ is significantly negative, indicates that Hypothesis 3a is validated. If $\omega_{2}$ is significantly positive, indicates that Hypothesis 3b is validated.

## 4. Results

### 4.1. Descriptive Statistics

Table 2 presents the mean, standard error, minimum, median, and maximum value of the main variables. Several points are worth noting. First, the mean of Ecsr is 0.506, the standard error is 1.541, and the maximum value is 5.962. The mean of Icsr is 4.893, the standard error is 7.378, and the maximum value is 21.193, indicating that there is a large gap between Ecsr and Icsr of each firm. Second, the mean of Nar is 4.472, the standard error is 0.402, and the maximum value is 5.295, a figure similar to Fung et al. [48], indicating that most chairmen in the sample tend to be narcissistic. Third, the mean of Ana is 1.655, the standard error is 1.003, and the maximum value is 3.872, which shows that different firms have different levels of analyst coverage. Fourth, the mean of Lev is 0.335, indicating that the sample firms have a relatively high asset-liability ratio. The mean of Dudb is 0.376, which complies with the requirement that the percentage of independent directors should not be less than 1/3. The mean of Con is 65.177, showing a high concentration of ownership in the sample companies. The mean of Gend is 0.950, which means 95.00% of the Gend in our sample are male. The mean of Lage is 52.249, the median is 51, which implies most chairmen are above 51 years old. The mean of Edu is 3.405, the median is 4, which implies most of the chairmen obtained bachelor’s and master’s degrees. Finally, other control variables are consistent with theoretical expectations.

Table 3 presents the correlations among relevant variables. All of the correlations between variables were smaller than 0.70 and were within acceptable limits. Chairman narcissism is significantly positively related with External CSR (β = 0.061, *p* < 0.05), but negatively correlated with Internal CSR (β = −0.075, *p* < 0.01). Size, Lev, Board is significantly positively related to External CSR and Internal CSR. On the contrary, Dudb, Edu is significantly positively related to External CSR, but negatively related to Internal CSR. Furthermore, we checked the variance of inflation factor (VIF) of all variables. Furthermore, we checked for the presence of multicollinearity and found that all variables had a VIF < 10, and the mean VIF for the main four models was <3. The Durbin–Watson (DW) test is a classic method for testing autocorrelation. We found that the Durbin–Watson statistics close to 2, which indicates that there is no autocorrelation in the model and no correlation between the sample data.

### 4.2. Regression Results

Table 4 presents the results of Generalized Estimating Equations (GEE) model to investigate the effect of chairman narcissism on external CSR and internal CSR and the moderating effect of Ana (Analyst coverage). The sample firms have multiple observations in different time dimensions, which are consistent with the characteristics of longitudinal data. A GEE model can handle the non-independence of repeated measures data and derive the maximum likelihood estimates. The dependent variable in Models 1–3 is external CSR, and Models 4–6 is Internal CSR. Model 1 and 4 contains only control variables, Model 2 and 5 introduce the independent variable, Models 3 and 6 introduce the moderating variables. Specifically, Models 1 and 2 show the direct of chairman narcissism on external CSR. The effect of chairman narcissism on external CSR is positive and significant (β = 0.227, *p* < 0.05 in Model 2), Hypothesis 1 is supported. Models 4 and 5 show the direct of chairman narcissism on Internal CSR. The effect of chairman narcissism on Internal CSR is negative and significant (β = −1.276, *p* < 0.05 in Model 5), Hypothesis 2 is supported.

Model 3 examines the interaction effect of chairman narcissism and Ana on the external CSR. Chairman narcissism has a positive and significant coefficient (β = 3.264, *p* < 0.05), and Ana has a positive and significant coefficient (β = 5.981, *p* < 0.05), which are broadly consistent with the results of Hong et al. [18]. However, the coefficient of the interaction between Ana and chairman narcissism is negative and significant (β = −1.366, *p* < 0.05), indicating that Ana weakens the positive relationship between chairman narcissism and external CSR. Hypothesis 3a is supported. Model 6 examines the interaction effect of chairman narcissism and Ana on the Internal CSR. Chairman narcissism has a negative and significant coefficient (β = −36.110, *p* < 0.01), and Ana has a negative and significant coefficient (β = −65.430, *p* < 0.01). The coefficient of the interaction between Ana and chairman narcissism is positive and significant (β = 15.636, *p* < 0.01), indicating that Ana strengthens the negative relationship between chairman narcissism and Internal CSR. Hypothesis 3b is supported.

### 4.3. Robustness Tests

We took several steps to ensure robustness in reported results for the effect of chairman narcissism on CSR (Icsr and Ecsr) and the moderating effect of Ana. First, we used external CSR and internal CSR one year later (i.e., ECSRt + 1, ICSRt + 1) as the dependent variable and re-ran the chairman on it. Models 1–4 in Table 5 present the additional analysis. They are similar to those reported above, indicating that chairman narcissism has a positive effect on external CSR and has a negative effect on Internal CSR. Ana negatively moderates the relationship between chairman narcissism and external CSR and positively moderates the relationship between chairman narcissism and Internal CSR. Our results support Hypothesis 1, Hypothesis 2, Hypothesis 3a, and Hypothesis 3b.

Second, we made a hypothetical scenario to measure chairman narcissism as a dummy variable. The measure then becomes as follows: Chairman narcissism (dummy) = 1 if chairman narcissism > 4.472 (Mean), chairman narcissism (dummy) = 0 if chairman narcissism < 4.472 (Mean). Models 5–8 in Table 5 present the regression results for the effect of the dummy variables of narcissism on the CSR (external CSR and Internal CSR) and the moderating effect of Ana. As shown in Model 5, the coefficient of chairman narcissism is consistently significant and positive, supporting Hypothesis 1. In Model 7, the coefficient of chairman narcissism is consistently significant and negative, supporting Hypothesis 2. In Model 6, the coefficient of the interaction between Ana and chairman narcissism is negative and significant, supporting Hypothesis 3a. In Model 8, the coefficient of the interaction between Ana and chairman narcissism is positive and significant, supporting Hypothesis 3b.

Third, to address the endogeneity between chairman narcissism and CSR, we use a logistic regression to test all hypotheses. We operationalized external CSR and internal CSR as dummy variable that equals 1 if the firm made external CSR and internal CSR, 0 otherwise. The results of the logistic regression analysis are reported in Table 6. Model 1 and 2 show the regression result for the external CSR dummy as the dependent variable. The result shows the effect of the estimated chairman narcissism on external CSR dummy, which is positive and significant at the 10% level (β = 0.525, *p* < 0.10), supporting Hypothesis 1, and the interaction between Ana and chairman narcissism is negative and significant (β = −6.107, *p* < 0.01), supporting Hypothesis 3a. Models 3 and 4 show the regression result for the internal CSR dummy as the dependent variable. The result shows the effect of the estimated chairman narcissism on internal CSR dummy, which is negative and significant at the 5% level (β = −0.380, *p* < 0.05), supporting Hypothesis 2, and the interaction between Ana and chairman narcissism is positive and significant (β = 6.834, *p* < 0.01), supporting Hypothesis 3b.

## 5. Discussion

### 5.1. Theoretical Implications

We make three contributions to the extant literature. First, our study enriches the relationship between chairman narcissism and CSR. Previous studies have explored the relationship between narcissistic leadership and CSR. However, research on this topic ignored the effect of chairman narcissism on different dimensions of CSR in China, an important emerging economy. Our study found that chairman narcissism has different effects on internal CSR and external CSR. This suggests that driven by satisfying self-interest, narcissistic chairmen exhibit different CSR practices when they treat stakeholders with different relationships to themselves. The conclusion clarifies the relationship between narcissism and CSR and provides a new perspective for related research. Second, our study broadens the understanding of the boundary conditions of narcissistic outcomes for chairman. Previous studies have explored the moderating effects of CEO duality [54], Public Attention [55], and Responsibility Experiences [56], but the impact of analyst coverage still needs to be examined. This study shows that analyst reporting is an essential factor that affects the relationship between narcissism and CSR. The conclusion implies that the effect of chairman narcissism on CSR will be affected by moderating variables, and subsequent studies should be differentiated. Third, our study expands the upper echelons theory. Previous studies have primarily examined the effects of the demographics of a corporate chairman, such as CEO duality [57], marital status [58], political connections [59], age (generation) [60], and poverty experience [61], on the firm’s CSR. Although the relevant research results are abundant, adopting demographic characteristics to reflect senior leaders’ psychological and cognitive characteristics has unreasonable operational methods and makes it challenging to explore the psychological processes of senior leaders in CSR practices. Narcissism, a fundamental personality dimension, could influence the firm’s CSR activities. Our study focused on chairman narcissism and examined the impact of chairman narcissistic personality traits on internal CSR and external CSR, which will expand the upper echelons theory.

### 5.2. Practical Implications

Our findings have three practical implications for managers and policymakers. First, the CSR behavior of the narcissistic chairman should be reasonably recognized. When companies recruit top executives, they should examine work experience, competence structure, and social connections, and also pay attention to the chairman’s personality traits. At the same time, Chinese firms must be cautious about the “high-profile” external CSR behavior of narcissistic chairman and avoid the negative impact on internal stakeholders. Second, companies should improve external governance mechanisms. This study shows that analyst coverage can improve corporate governance quality and weaken the impact of chairman narcissism on internal and external CSR. Shareholders can guide analysts to report production and operation information of enterprises to improve the authenticity, credibility, and decision-making quality of information and thus protect their own interests. External investors can use analyst coverage and research reports as important reference indicators to evaluate the risk of corporate violations when examining investment targets. Third, enterprises should balance the social responsibilities of internal and external stakeholders. Under limited resources, enterprises can increase external CSR investment to obtain external resources but should not neglect the investment in internal employee welfare. External CSR has a public relations effect and becomes a tool to divert public attention. However, employees are the real wealth of an enterprise. Enterprises should recognize the role of employees in the development of the enterprise and achieve a win-win situation by establishing a staff welfare system.

### 5.3. Limitations and Future Research

Although this paper examines the effect of chairman narcissism on CSR and the moderating effect of analyst reports, there are still many limitations to this paper that require subsequent improvements. First, the video survey method to measure chairman narcissism has been supported by the literature [21,48], but the professional knowledge and subjective attitude of the raters in the survey method will affect the results and lead to biased results. Subsequent studies should adopt more reasonable and comprehensive indicators to measure chairman narcissism. Second, this paper examines the effect of chairman narcissism on internal and external CSR but does not examine the economic consequences of the two types of CSR. Future research could explore how chairman narcissism affects corporate performance by influencing internal and external CSR.

## 6. Conclusions

How chairman narcissism affects corporate behavior is essential in studying organizational behavior. Based on the data of Shanghai and Shenzhen A-share listed companies from 2009–2019, this study examines the relationship between chairman narcissism and external CSR and internal CSR using the upper echelons theory and stakeholder theory. We show that chairman narcissism has a positive and significant effect on external CSR, has a negative and significant effect on internal CSR. The reason is that, driven by narcissistic traits, chairmen treat stakeholders with different relationships and hold different behavioral logic. On the one hand, to gain continuous attention and praise from external stakeholders, narcissistic chairmen will actively undertake external CSR. On the other hand, the negative traits of narcissistic chairmen, such as egoism and intellectual inhibition, drive them to treat internal stakeholders negatively or even sacrifice the interests of internal stakeholders to maximize their interests. Further study found that analyst coverage weakens the effects of chairman narcissism on internal and external CSR. This indicates that as an external governance mechanism of the capital market, analyst coverage can exert the “information disclosure effect” and “supervision effect” to reduce the irrational donation of the narcissistic chairman and excessive inhibition of employee behavior.

## Figures and Tables

**Table 1 behavsci-13-00245-t001:** Sixteen-item narcissistic personality index of chairman.

**Dimensions**	**Title Item**
Entitlement/Exploitativeness	He/she usually dominates all conversations.
	People always seem to recognize his/her authority.
	He/she has a strong will to power.
	He/she likes to manipulate people.
Authority/Leadership	He/she has a high reputation in the industry.
	He/she likes people to obey his/her.
	He/she is more capable than others.
	He/she likes to establish authority over others.
	He/she likes to be followed.
Superiority/Arrogance	He/she is not willing to listen to subordinates.
	He/she is extraordinary and exceptional.
	He/she always knows what he/she is doing.
Self-admiration/Self-absorption	He/she always likes to be the center of attention.
	He/she sometimes gets angry when people complain about him/her.
	He/she is willing to express his/her opinion in public.
	He/she can usually be convinced to solve any problem.

**Table 2 behavsci-13-00245-t002:** Main descriptive statistics.

**Variable**	**Sample Size**	**Mean**	**SD**	**Min**	**Med**	**Max**
Ecsr	1212	0.506	1.541	0	0	5.962
Icsr	1212	4.893	7.378	0	0	21.193
Nar	1212	4.472	0.402	3.500	4.521	5.292
Ana	1212	1.655	1.003	0	1.892	3.872
Size	1212	21.943	1.185	19.833	21.716	26.250
Age	1212	2.591	0.410	1.099	2.639	3.526
Lev	1212	0.336	0.183	0.027	0.314	0.897
Dudb	1212	0.376	0.055	0.300	0.333	0.571
Sub	1212	16.081	1.795	7.601	16.185	21.123
Board	1212	2.127	0.179	1.609	2.197	2.708
Con	1212	65.177	14.160	23.620	67.610	100
Gend	1212	0.950	0.217	0	1	1
Lage	1212	52.249	6.729	33	51	76
Edu	1212	3.405	1.038	1	4	5

**Table 3 behavsci-13-00245-t003:** Main Variables Correlation Matrix.

	**Ecsr**	**Icsr**	**Nar**	**Ana**	**Size**	**Age**	**Lev**	**Dudb**	**Sub**	**Board**	**Con**	**Gend**	**Lage**	**Edu**
Ecsr	1													
Icsr	0.01	1												
Nar	0.061 **	−0.075 ***	1											
Ana	−0.01	0.091 ***	0.061 **	1										
Size	0.363 ***	0.062 **	0.027	0.016	1									
Age	−0.015	0.02	0.087 ***	0.001	0.117 ***	1								
Lev	0.191 ***	0.110 ***	0.053 *	0.006	0.627 ***	0.202 ***	1							
Dudb	0.069 **	−0.094 ***	0.069 **	−0.064 **	0.074 **	−0.026	−0.005	1						
Sub	0.201 ***	−0.006	0.019	−0.051 *	0.374 ***	−0.079 ***	0.150 ***	0.024	1					
Board	0.091 ***	0.054 *	−0.066 **	0.045	0.082 ***	0.050 *	0.056 *	−0.642 ***	0.059 **	1				
Con	0.075 ***	−0.032	0.042	0.046	0.061 **	−0.230 ***	−0.100 ***	0.105 ***	0.043	−0.091 ***	1			
Gend	−0.024	−0.017	0.015	0.179 ***	0.057 **	−0.028	0.079 ***	−0.091 ***	0.041	0.035	−0.128 ***	1		
Lage	−0.006	−0.02	0.051 *	−0.001	0.160 ***	0.222 ***	0.068 **	−0.113 ***	−0.069 **	0.144 ***	−0.086 ***	0.042	1	
Edu	0.092 ***	−0.099 ***	−0.029	−0.066 **	0.147 ***	0.005	0.01	0	0.123 ***	0.028	−0.164 ***	0.108 ***	−0.184 ***	1

Note: * *p* < 0.10, ** *p* < 0.05, *** *p* < 0.01.

**Table 4 behavsci-13-00245-t004:** Regression Results.

**Variables**	**Ecsr**			**Icsr**		
	**Model 1**	**Model 2**	**Model 3**	**Model 4**	**Model 5**	**Model 6**
Size	0.443 ***	0.447 ***	0.444 ***	0.383	0.361	0.358
	(8.74)	(8.83)	(8.77)	(1.49)	(1.41)	(1.41)
Age	−0.122	−0.142	−0.114	−0.001	0.113	−0.232
	(−1.14)	(−1.33)	(−1.06)	(−0.00)	(0.21)	(−0.43)
Lev	−0.195	−0.228	−0.190	2.839 *	3.019 **	2.715 *
	(−0.65)	(−0.76)	(−0.63)	(1.87)	(1.99)	(1.81)
Dudb	3.627 ***	3.548 ***	3.407 ***	−15.028 ***	−14.585 ***	−12.524 **
	(3.70)	(3.62)	(3.47)	(−3.02)	(−2.94)	(−2.55)
Sub	0.047 *	0.045 *	0.047 *	−0.099	−0.088	−0.091
	(1.87)	(1.80)	(1.88)	(−0.78)	(−0.70)	(−0.73)
Board	1.345 ***	1.368 ***	1.342 ***	−0.724	−0.848	−0.630
	(4.47)	(4.55)	(4.47)	(−0.47)	(−0.56)	(−0.42)
Con	0.004	0.004	0.004	−0.025	−0.022	−0.026 *
	(1.41)	(1.27)	(1.31)	(−1.59)	(−1.43)	(−1.66)
Gend	−0.249	−0.260	−0.202	−0.900	−0.835	−1.950 **
	(−1.30)	(−1.36)	(−1.04)	(−0.93)	(−0.86)	(−2.00)
Lage	−0.010	−0.011 *	−0.011 *	−0.083 **	−0.079 **	−0.072 **
	(−1.57)	(−1.67)	(−1.73)	(−2.49)	(−2.38)	(−2.18)
Edu	0.050	0.051	0.052	−0.844 ***	−0.849 ***	−0.821 ***
	(1.18)	(1.21)	(1.24)	(−3.97)	(−4.00)	(−3.91)
Nar		0.227 **	3.264 **		−1.276 **	−36.110 ***
		(2.23)	(2.26)		(−2.47)	(−4.99)
Ana			5.981 **			−65.430 ***
			(2.07)			(−4.53)
Nar×Ana			−1.366 **			15.636 ***
			(−2.11)			(4.82)
_cons	−13.492 ***	−14.442 ***	−27.737 ***	13.954 **	19.292 ***	165.712 ***
	(−10.93)	(−11.08)	(−4.25)	(2.23)	(2.92)	(5.07)
N	1212	1212	1212	1212	1212	1212
Chi2	233.027	238.957	244.384	45.166	51.516	86.051

Note: * *p* < 0.10, ** *p* < 0.05, *** *p* < 0.01.

**Table 5 behavsci-13-00245-t005:** Robustness Tests.

**Variables**	**Ecsr**		**Icsr**		**Ecsr**		**Icsr**	
	**Model 1**	**Model 2**	**Model 3**	**Model 4**	**Model 5**	**Model 6**	**Model 7**	**Model 8**
Size	0.393 ***	0.393 ***	0.451 *	0.447 *	0.446 ***	0.445 ***	0.421	0.388
	(8.29)	(8.29)	(1.82)	(1.82)	(8.79)	(8.79)	(1.62)	(1.51)
Age	−0.244 **	−0.218 **	0.732	0.423	−0.127	−0.097	0.007	−0.242
	(−2.45)	(−2.17)	(1.40)	(0.81)	(−1.19)	(−0.91)	(0.01)	(−0.45)
Lev	−0.070	−0.044	4.340 ***	4.073 ***	−0.219	−0.195	3.135 **	3.114 **
	(−0.25)	(−0.16)	(2.97)	(2.81)	(−0.73)	(−0.65)	(2.05)	(2.05)
Dudb	4.333 ***	4.178 ***	−12.925 ***	−11.066 **	3.569 ***	3.489 ***	−14.760 ***	−13.671 ***
	(4.72)	(4.55)	(−2.70)	(−2.33)	(3.64)	(3.56)	(−2.94)	(−2.74)
Sub	0.071 ***	0.071 ***	−0.047	−0.049	0.047 *	0.050 **	−0.115	−0.108
	(3.04)	(3.06)	(−0.38)	(−0.40)	(1.90)	(1.99)	(−0.90)	(−0.85)
Board	1.547 ***	1.529 ***	−1.816	−1.623	1.357 ***	1.326 ***	−0.777	−0.635
	(5.49)	(5.43)	(−1.23)	(−1.11)	(4.51)	(4.41)	(−0.50)	(−0.42)
Con	0.001	0.001	−0.046 ***	−0.049 ***	0.004	0.004	−0.021	−0.024
	(0.19)	(0.27)	(−3.02)	(−3.25)	(1.28)	(1.32)	(−1.30)	(−1.51)
Gend	−0.248	−0.167	−0.762	−1.776 *	−0.272	−0.224	−0.660	−1.518
	(−1.39)	(−0.91)	(−0.82)	(−1.88)	(−1.42)	(−1.15)	(−0.67)	(−1.53)
Lage	−0.012 **	−0.013 **	−0.056 *	−0.049	−0.011	−0.010	−0.077 **	−0.076 **
	(−1.99)	(−2.08)	(−1.74)	(−1.54)	(−1.64)	(−1.57)	(−2.29)	(−2.27)
Edu	0.066 *	0.065 *	−0.852 ***	−0.826 ***	0.050	0.053	−0.890 ***	−0.863 ***
	(1.69)	(1.65)	(−4.15)	(−4.06)	(1.20)	(1.27)	(−4.15)	(−4.04)
Nar	0.266 ***	3.011 **	−1.382 ***	−32.536 ***	0.147 *	2.780 **	−1.200 ***	−20.213 ***
	(2.79)	(2.23)	(−2.78)	(−4.65)	(1.78)	(2.56)	(−2.84)	(−3.67)
Ana		5.203 *		−58.400 ***		0.554		−0.529
		(1.93)		(−4.18)		(1.56)		(−0.29)
Nar×Ana		−1.232 **		13.983 ***		−1.189 **		8.557 ***
		(−2.03)		(4.45)		(−2.43)		(3.44)
_cons	−14.160 ***	−25.788 ***	16.291 **	147.018 ***	−13.549 ***	−14.866 ***	13.343 **	15.910 **
	(−11.60)	(−4.22)	(2.56)	(4.65)	(−10.99)	(−10.23)	(2.11)	(2.15)
N	1212	1212	1212	1212	1212	1212	1212	1212
Chi2	272.175	278.783	76.956	107.619	236.800	243.979	56.087	78.901

Note: * *p* < 0.10, ** *p* < 0.05, *** *p* < 0.01.

**Table 6 behavsci-13-00245-t006:** Robustness Tests.

**Variables**	**Ecsr**		**Icsr**	
	**Model 1**	**Model 2**	**Model 3**	**Model 4**
Size	0.718 ***	0.755 ***	0.070	0.070
	(5.65)	(5.81)	(0.88)	(0.87)
Age	−0.440 *	−0.376	0.077	−0.056
	(−1.66)	(−1.41)	(0.46)	(−0.32)
Lev	−0.111	−0.045	0.783 *	0.711
	(−0.15)	(−0.06)	(1.69)	(1.50)
Dudb	2.195	1.820	−4.526 ***	−3.967 **
	(0.94)	(0.75)	(−2.77)	(−2.39)
Sub	0.097	0.109 *	−0.038	−0.034
	(1.57)	(1.75)	(−0.99)	(−0.88)
Board	1.569 **	1.435 **	−0.116	−0.070
	(2.35)	(2.08)	(−0.24)	(−0.14)
Con	−0.003	−0.002	−0.007	−0.008
	(−0.34)	(−0.29)	(−1.39)	(−1.63)
Gend	−0.723	−0.527	−0.298	−0.694 **
	(−1.62)	(−1.14)	(−1.03)	(−2.21)
Lage	−0.028	−0.029	−0.028 ***	−0.025 **
	(−1.54)	(−1.56)	(−2.66)	(−2.28)
Edu	0.084	0.099	−0.251 ***	−0.258 ***
	(0.72)	(0.82)	(−3.82)	(−3.85)
Ana		27.076 ***		−28.236 ***
		(2.95)		(−5.13)
Nar	0.525 *	14.158 ***	−0.380 **	−15.733 ***
	(1.86)	(3.17)	(−2.41)	(−5.44)
Nar×Ana		−6.107 ***		6.834 ***
		(−3.07)		(5.33)
_cons	−23.178 ***	−84.666 ***	4.467 **	68.193 ***
	(−7.74)	(−4.02)	(2.16)	(5.44)
N	1212	1212	1212	1212
Chi2	139.044	151.458	46.783	90.332

Note: * *p* < 0.10, ** *p* < 0.05, *** *p* < 0.01.

## Data Availability

The data that support the findings of this study are available from the corresponding author (Q.G.) upon reasonable request.

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
