# Peer review of "Chairman Narcissism and Social Responsibility Choices: The Moderating Role of Analyst Coverage"

_behavsci, 2023, doi:10.3390/bs13030245_

Round 1

Reviewer 1 Report

The topic is interesting and timely; however, there are several drawbacks that should be corrected prior the publication. They are as follows:

1. Intoridction is quite informative; however, the authors should state the research aim/goal in it. I.e. they should explain what the research was conducted for.

2. The scientific gap should be highlighted in the introduction as well, i.e. the authors ought to explain which scientific knowledge they contribute to.

3. Literature review is well-written.

4. In the methodology, the authors should explain how they calculated the necessary sample size as, at present, it is not obvious that their study is representative.

5. The authors use regression analysis for their study, which could be used for such research. They found that there was no multicollinearity; however, I failed to find any information on autocorrelation. The authors should apply additional (e.g. Durbin-Watson) test.

6. The Rare very low, i.e. it means that the proposed regression models do not fit well. Maybe, the authors could apply different methods or explain how they deal with low R2.

7. Discussion and conclusions should be in different parts. In the discussion, the authors should compare their obtained research results with other similar research while in conclusions they should provide a summary of the outcomes.

8/ Please, consider citing "Differences in Attitude to Corporate Social Responsibility Among Generations" (https://www.mdpi.com/2071-1050/13/19/10944) 

Author Response

Response to Reviewer 1 Comments

Point 1: Intoridction is quite informative; however, the authors should state the research aim/goal in it. I.e. they should explain what the research was conducted for.

Response 1: Thank you for your careful evaluation of this manuscript. We greatly appreciate the valuable comments and helpful suggestions. As suggested, we have enhanced the discussion of the research aim/goal, and the paper is revised accordingly. Revised section at lines 63-65. The main purpose of the study is to investigate the effect of chairman narcissism on various dimensions of CSR and the moderating role of the analyst coverages. Our logic for the research aim was developed in the following manner. First, summarized the research aim/goal; Second, explain the progress of this paper towards this goal.

Point 2: The scientific gap should be highlighted in the introduction as well, i.e. the authors ought to explain which scientific knowledge they contribute to..

Response 2: Thank you for this useful comment. As suggested, we conclude by summarizing the contributions on chairman narcissism, on internal and external CSR. Revised section at lines 77-91. This paper contributions to the extent literature on chairman narcissism and on internal CSR, external CSR in several ways. First, this study contributes to the growing literature on the relationship between chairman narcissism and the choice of CSR strategies. Our results show that chairman personality traits, especially narcissism, can significantly impact the choice of a firm’s nonmarket strategies in China, an important emerging economy. Compiling a chairman narcissism index from a video survey, we show that firms with narcissistic chairmen engage in more external CSR activities and less internal CSR activities. We provide a more comprehensive assess-ment of the relationship between chairman narcissism and CSR strategies, and clarified the paradoxical relationship between chairman narcissism and CSR. Second, we respond to the growing calls for deconstructing the CSR measure. Our study distinguished CSR into internal CSR and external CSR, this division extending and enriching the literature of internal CSR and external CSR. More importantly, we expand the theory of stakeholder. Meanwhile, by relating chairmen’ psychological characteristics such as nar-cissism to explain CSR choices, we open up doors for future research on the discre-tionary determinants of CSR. Third, this paper broadens the boundaries of the relationship between chair-man narcissism and CSR. Based on the call, narcissistic traits should be studied in the context of “activating” and “inhibiting”. We choose analyst coverage as the mod-erator variable and the result show that analyst coverage weakened the relationship between chairman narcissism and CSR, which enrich the contingency perspective of chairman narcissism.

Point 3: Literature review is well-written.

Response 3: Thank you for your comment. We further enrich the literature review. First, revised section at lines 99. Narcissism is a fundamental personality trait that refers to an individual's self-worship, self-superiority, lack of empathy, selfishness, and pursuit of authority, power, and exploitation. Second, revised section at lines 110-113. In terms of motivation, narcissistic chairmen are motivated to seek power, control and inflated ego, especially motivated by various types of behavior inviting applause and admiration[19, 21]. Third, revised section at lines 116. Stakeholder theory argues that the firm can be con-sidered as a collection of contracts composed of stakeholders. Fourth, Drawing upon stakeholder theory. Fifth, revised section at lines 164. Stakeholder theory shows that the company can be regarded as a collection of contractual relationships. Sixth, revised section at lines 179-180. At the same time, narcissism chairman also exhibits higher vulnerability, including distrust, deceitfulness, oppositionality, callousness and anxiety [37, 38].

Point 4: In the methodology, the authors should explain how they calculated the necessary sample size as, at present, it is not obvious that their study is representative.

Response 4: Thank you for this useful comment. As suggested, we have calculated the sample size, especially for the narcissistic chairmen. Revised section at lines 249-252. Data on corporate chairman narcissism were collected from a video survey methodology, and videos of chairmen in our sample through publicly available internet sources, such as Baidu.com and hao.360.com search engines. The name of the firm and chairman’s name from the Chinese Stock Market Research (CSMAR) database. CSMAR is a leading and professional financial database of Chinese listed firms. Revised section at lines 275-276. Following Petrenko et al. [21], Fung et al. [49], Zhu and Chen. [50], this paper uses a video survey method to study chairman narcissism. This section following Petrenko et al. [21], Fung et al. [49], Zhu and Chen. [50] to enhance the representativeness of the research. Revised section at lines 300-308. On the other hand, we created an excel spreadsheet consisting of the NPI-16 scales and divided the 12 raters into 6 groups. The raters were required to enter the scoring system with an individual login password, watch the video independently within the specified time, and score the narcissism degree of each chairman. And the raters also required to enter their ratings and justifications for each chairman’s rating in the excel spreadsheet. The justifications described the reasons why the rater gave a particular score to a chairman [51]. Specific justifications included whether a chairman used friendly facial expressions, exhibited body posture, spoke casually, and wore colorful clothing [49].

Point 5: The authors use regression analysis for their study, which could be used for such research. They found that there was no multicollinearity; however, I failed to find any information on autocorrelation. The authors should apply additional (e.g. Durbin-Watson) test.

Response 5: Thank you for this useful comment. As suggested, we added autocorrelation tests and provided test results. Revised section at lines 388-391. The Durbin-Watson (DW) test is a classic method for testing autocorrelation. We found that the Durbin Watson statistics close to 2, which indicates that there is no au-to-correlation in the model and no correlation between the sample data.

Point 6: The R2 are very low, i.e. it means that the proposed regression models do not fit well. Maybe, the authors could apply different methods or explain how they deal with low R2.

Response 6: Thank you for this useful comment. As suggested, we used the Generalized Estimating Equations (GEE) model to investigate the effect of chairman narcissism on external CSR and internal CSR and the moderating effect of Ana (Analyst coverage). Revised section at lines 398-401. The sample firms have multiple observations in different time dimensions, which are consistent with the characteristics of longitudinal data. A GEE model can handle the non-independence of repeated measures data and derive the maximum likelihood es-timates. Meanwile, the regression results are shown in Table 4. Revised section at lines 422.

Point 7: Discussion and conclusions should be in different parts. In the discussion, the authors should compare their obtained research results with other similar research while in conclusions they should provide a summary of the outcomes.

Response 7: Thank you for this useful comment. As suggested, discussion and conclusions should be in different parts. We have placed the conclusion in chapter 6. Revised section at lines 524. Meanwhile , we have revised the discussion section, revised section at lines 467-470. Previous studies have explored the relationship between narcissistic leadership and CSR. But research on this topic ignored the effect of chairman narcissism on different dimensions of CSR in China, an important emerging economy.

Point 8: Please, consider citing "Differences in Attitude to Corporate Social Responsibility Among Generations" (https://www.mdpi.com/2071-1050/13/19/10944).

Response 8: Thank you for this useful comment. As suggested, we have cited《Differences in Attitude to Corporate Social Responsibility among Generations》. Revised section at lines 688-689.

Aattachment is the revised article, please open it in Microsoft Word.

Reviewer 2 Report

Key words: Corporate social responsibility (CSR); CSR strategies; Behavioral consistency theory; personality traits; narcissistic chairman; management autonomy; cognitive and motivational factors.  

This paper clarifies the features of narcissistic personalities, especially in the field of Corporate social responsibility (CSR) firms and the related management of power by the sides of CSR chairmen. As affirmed by authors the perspective with which internal and external stakeholders perceive the firm respectively is different and somehow opposite: the ones take care of their interests, while the others emphasize more the role social ethics and norms should have in the firm. In Section 2 the authors state their Hypothesis 1, that is “Chairman narcissism positively influences the external CSR.” (see line 151). The authors try to illustrate this CSR firm landscape with a chairman’s role as an internal stakeholder behaves on the interest of the company. Features of a narcissistic self-centered personality then emerges especially at lines 89‒121, as one asking for an exclusive attention and the search of economic benefits in the short term. This “positively affects external stakeholders” and “negatively affects the internal CSR.”

Features of a narcissistic self-centered personality are then reinforced at lines 153‒185 in the context of the Internal CSR. (Hypothesis 2) according to authors (see line 186). Narcissistic traits of chairmen are considered under a quite exclusive binary angle: i. the activation of good behavior by one side, and ii. the inhibition of behavior by the other. Since the authors’ target is “to motivate the narcissistic chairmen to make better internal and external CSR decisions based on the interests of shareholders,” they formulate a double Hypothesis 3a and 3b: “Analyst coverage weakens the positive relationship between narcissism and external CSR,” and “Analyst coverage weakens the negative relationship between chairman narcissism and internal CSR.” (see lines 223-226). This means that Ana (Analyst coverage) has an “information disclosure effect,” in the sense that it reduces “information asymmetry between CSR inputs and stakeholders,” improves “information transparency, and reduce the impact of irrational narcissistic traits of the chairman on external and internal CSR.” (especially lines 188‒197, and 198‒201 for quotation marks).  

In Section 3 Methods, the authors explain their sample includes Chinese A-share listed firms in Shanghai and Shenzhen stock exchanges from 2010 to 2019. The final sample comprises 1211 firm-year observations related to 169 firms. As affirmed by authors, “Based on the Chinese institutional context, the chairman of the board is the most powerful individual in the managerial team, directly influencing CSR decisions and behaviors.” (see lines 254-56). The authors then measure the narcissistic personality of chairmen through an inventory developed by Raskin and Hall in 1981, and modified by Emmons in 1984, based on 16 indicators (for an illustration of indicators see Table 1 at line 295) related to 4 dimensions (Entitlement/exploitativeness; Autority/Leadership; Superiority/Arrogance; and Self-admiration/self-absorption). 12 psychology graduate students rated the replies by chairmen using also of videos collated by authors on chairmen’s activity.

Control variables were demographic profile of chairmen, and characteristics of firms such as size, age, and assets. Section 4. Results, shows some accurate descriptive statistics (see Table 2 at line 358, and Table 3 at line 369 for the Correlation Matrix) on the scoring obtained during the sample’s screening, indicate Chairman narcissism is significantly positively related with External CSR, but negatively correlated with Internal CSR. Variance of inflation factor (VIF) of all variables is also checked by authors. Regression results are clearly presented in Table 4, line 395. Construct and predictive validity were evaluated by examining relations between narcissism and relevant criteria observed on individual firms by two Robustness Tests are conducted at Table 5, line 433, and Table 6, line 438, respectively. They both confirm the hypotheses by authors on the impact of chairman narcissistic personality traits on internal and external CSR. Practical implications highlighted by authors are i. a more cautious attention by firms to the chairman’s personality traits, ii. the improvement of corporate governance quality, and iii. a better balancing of social responsibilities of internal and external stakeholders. Conclusions praise for an empowering of external analyst coverage by the side of external stakeholders so that to reduce the excessive negative influence of narcissistic personality traits of chairmen on internal employees’ behavior.

MINOR AMENDMENTS:

§  I personally would a little re-discuss the Raskin and Hall’s inventory’s quality, specifying this model dates to first years of 1980, and so that it could be a little old-fashioned. In addition, you seem to overestimate the grandiosity of a narcissistic personality without considering at all, or only in very small part, the vulnerability’s aspects of narcissistic personalities such as referenced in [1], [2], and [3] below,

§  I would rewrite the abstract, highlighting in a more punctual statistical fashion the work you have done along the paper with stats and related comments,

§  I personally would delete the following sentence: As the “information lubricant” of the capital market,… at line 297. It does make any sense to your interesting analysis.

References:

[1] Miller JD, Lynam DR, Hyatt CS, Campbell WK. Controversies in Narcissism. Annu Rev Clin Psychol. 2017 May 8;13:291-315. doi: 10.1146/annurev-clinpsy-032816-045244. Epub 2017 Mar 15. PMID: 28301765.

[2] Miller JD, Lynam DR, Campbell WK. Measures of Narcissism and Their Relations to DSM-5 Pathological Traits: A Critical Reappraisal. Assessment. 2016 Feb;23(1):3-9. doi: 10.1177/1073191114522909. Epub 2014 Feb 17. PMID: 24550548.

[3] Henttonen P, Salmi J, Peräkylä A, Krusemark EA. Grandiosity, vulnerability, and narcissistic fluctuation: Examining reliability, measurement invariance, and construct validity of four brief narcissism measures. Front Psychol. 2022 Oct 10;13:993663. doi: 10.3389/fpsyg.2022.993663. PMID: 36300061; PMCID: PMC9589046.

Kind Regards,

Author Response

Point 1: Comments and Suggestions for Authors. Key words: Corporate social responsibility (CSR); CSR strategies; Behavioral consistency theory; personality traits; narcissistic chairman; management autonomy; cognitive and motivational factors.

Response 1: Thank you for your careful evaluation of this manuscript. We greatly appreciate the valuable comments and helpful suggestions. As suggested, we have modified the keywords. Revised section at lines 27-28. Corporate social responsibility (CSR); CSR strategies; Behavioral consistency theory; personality traits; narcissistic chairman; management autonomy; cognitive and motivational factors.

Point 2: I personally would a little re-discuss the Raskin and Hall’s inventory’s quality, specifying this model dates to first years of 1980, and so that it could be a little old-fashioned. In addition, you seem to overestimate the grandiosity of a narcissistic personality without considering at all, or only in very small part, the vulnerability’s aspects of narcissistic personalities such as referenced in [1], [2], and [3] below.

Response 2: Thank you for this useful comment. As suggested, we discussed some of our research on chairman narcissism and revised it based on your comments. First, in our research, we followed the idea of the Raskinb and Hall, and his research used widely and representative in management research. In the actual research, we followed the Narcissism Personality Inventory (NPI) developed by Fung et al. [49], Qiao et al. [52], whose main research can measure the narcissistic chairman of  Chinese listed firms. In this revision, we cited the Narcissism Personality Inventory (NPI-16). Revised section at lines 279,283. Second, data on chairman narcissism were collected from a video survey methodology. The video approach, which is a method rapidly gaining acceptance and credibility in the literature [21,49,50], requires trained raters to watch a designated length of a video featuring a firm’s Board chair and give a numerical score on a set of items that measures narcissism, which reflects perceived narcissism by raters. The purpose of using the NPI-16 is to facilitate the raters rating of the chairman's narcissism. The “pre-measuring method” (N=30) is an important stage to examine the degree of narcissism (Grandiose Narcissism and Vulnerable Narcissism). However, we found the facial expressions, body postures, speaking styles, dressing styles of the narcissistic chairmen were not consistent with vulnerability narcissism(e.g., distrust, selfishness. deceitfulness, oppositionality, callousness, anxiety, depression, self-consciousness, and vulnerability). Therefore, table 1 has fewer items for vulnerability narcissism. Third, vulnerability narcissism has received attention from scholars. We draw on the research of Miller et al.[35], Miller et al.[37], and Henttonen et al.[38] to add a description of the narcissistic chairmen's vulnerability narcissism. Revised section at lines 171, distrusted, callousness. Revised section at lines 179-180. At the same time, narcissism chairman also exhibits higher vulnerability, including distrust, deceitfulness, oppositionality, callousness and anxiety [37, 38]. Fourth, in recent years, grandiose narcissism and vulnerable narcissism have received attention from scholars. Whether the narcissistic chairman of Chinese listed firms contain both grand narcissism and vulnerable narcissism will be the direction of our research. In future research, we will develop a Narcissism Personality Inventory consists of grand narcissism and vulnerable narcissism to examine the impact of chairman's grand narcissism and vulnerable narcissism on organizations.

Point 3: I would rewrite the abstract, highlighting in a more punctual statistical fashion the work you have done along the paper with stats and related comments.

Response 3: Thank you for your comment. I have rewrited the abstract. Revised section at lines 12-26. Chairman narcissism has received extensive attention in social psychology and organizational behavior, the relationship between chairman narcissism and social responsibility has not yet re-ceived much attention. The purpose of this study is to investigate the effect of chairman narcis-sism on various dimensions of CSR and the moderating roles of analyst coverages. Based on upper echelons theory and stakeholder theory, we distinguished internal corporate social re-sponsibility (internal CSR) and external corporate social responsibility (external CSR) according to whether there was a formal contractual relationship. This study used a narcissism index of chairmen of Chinese listed companies to examine the relationship between chairman narcissism and internal CSR, external CSR, and the data were analyzed using Stata16.0. The results showed that there was a positive correlation between chairman narcissism and external CSR, and there was a negative correlation between chairman narcissism and internal CSR. That is, the higher the Chairman’s narcissism degree is, the more external CSR and less internal CSR the firm makes. Further research showed that analyst coverage has weakened the impact of chairman narcissism on internal and external CSR. This paper enriches and expands the research on chairman narcis-sism and CSR, and provides new ideas for selecting corporate managers and improving corporate governance.

Point 4: I personally would delete the following sentence: As the “information lubricant” of the capital market,… at line 297. It does make any sense to your interesting analysis.

Response 4: Thank you for this useful comment. As suggested, I haved deleted the following sentence: As the “information lubricant” of the capital market. Revised section at lines 320.

Aattachment is the revised article, please open it in Microsoft Word.
